

# A variant-informed decision support system for tackling COVID-19: a transfer learning and multi-attribute decision-making approach

Amirreza Salehi Amiri[1], Ardavan Babaei[2], Vladimir Simic[3,4] and Erfan Babaee Tirkolaee[2,5,6]

[1] Department of Industrial Engineering, Sharif University of Technology, Tehran, Iran
[2] Department of Industrial Engineering, Istinye University, Istanbul, Turkey
[3] Faculty of Transport and Traffic Engineering, University of Belgrade, Belgrade, Serbia
[4] Department of Computer Science and Engineering, Korea University, Seul, Republic of Korea
[5] Department of Industrial Engineering and Management, Yuan Ze University, Taoyuan, Taiwan
[6] Department of Mechanics and Mathematics, Western Caspian University, Baku, Azerbaijan

Corresponding author
Erfan Babaee Tirkolaee,
erfan.babaee@istinye.edu.tr

## ABSTRACT

The global impact of the COVID-19 pandemic, characterized by its extensive societal, economic, and environmental challenges, escalated with the emergence of variants of concern (VOCs) in 2020. Governments, grappling with the unpredictable evolution of VOCs, faced the need for agile decision support systems to safeguard nations effectively. This article introduces the Variant-Informed Decision Support System (VIDSS), designed to dynamically adapt to each variant of concern's unique characteristics. Utilizing multi-attribute decision-making (MADM) techniques, VIDSS assesses a country's performance by considering improvements relative to its past state and comparing it with others. The study incorporates transfer learning, leveraging insights from forecast models of previous VOCs to enhance predictions for future variants. This proactive approach harnesses historical data, contributing to more accurate forecasting amid evolving COVID-19 challenges. Results reveal that the VIDSS framework, through rigorous K-fold cross-validation, achieves robust predictive accuracy, with neural network models significantly benefiting from transfer learning. The proposed hybrid MADM approach integrated approaches yield insightful scores for each country, highlighting positive and negative criteria influencing COVID-19 spread. Additionally, feature importance, illustrated through SHAP plots, varies across variants, underscoring the evolving nature of the pandemic. Notably, vaccination rates, intensive care unit (ICU) patient numbers, and weekly hospital admissions consistently emerge as critical features, guiding effective pandemic responses. These findings demonstrate that leveraging past VOC data significantly improves future variant predictions, offering valuable insights for policymakers to optimize strategies and allocate resources effectively. VIDSS thus stands as a pivotal tool in navigating the complexities of COVID-19, providing dynamic, data-driven decision support in a continually evolving landscape.

## INTRODUCTION

The COVID-19 pandemic, which emerged in 2020, rapidly spread globally and posed a significant health risk due to its extensive reach and prolonged duration. By the end of February 2024, there were 7,003,577 reported deaths worldwide, along with 703,875,382 confirmed cases (*Worldometer, 2023*). Furthermore, the pandemic has led to various challenges, significantly impacting society, the economy, and the environment (*Mete et al., 2023*). During the initial phase of the epidemic, it was crucial to comprehensively understand how the disease spreads and evolves. This understanding enables relevant authorities and groups to make informed decisions and implement appropriate preventative actions (*Naeem et al., 2021*). In 2020, the emergence of specific variants posed a global health threat, leading the World Health Organization (WHO) to classify them as variants of concern (VOCs). These VOCs were subject to prioritized global monitoring, research, and adaptive responses. According to the latest WHO definition, VOCs exhibit genetic changes that impact virus characteristics and demonstrate a growth advantage in at least one WHO region. Currently, the WHO recognizes five VOCs: Alpha, Beta, Gamma, Delta, and Omicron (*World Health Organization, 2023*; *Xia et al., 2024*).

Figure 1 illustrates the temporal evolution of nationwide VOC trends, specifically highlighting the VOCs of COVID-19, spanning from December 2021 to February 2024 (*BIOBOT, 2024*).

Governments grappled with uncertainty and volatility during the pandemic as these VOCs continued to evolve, challenging their efforts to protect their nations (*Thaker et al., 2023*). Due to the dynamic COVID-19 spread patterns, countries must establish efficient decision support systems that adapt to each VOC. These systems must adapt swiftly to changing situations and effectively address the challenges posed by the pandemic. As nations implement diverse policies to combat the crisis, outcomes vary significantly. An efficient decision support system can play a pivotal role by forecasting future spread trends, enabling proactive planning, resource allocation, and strategic economic management (*Ayris et al., 2022*). Additionally, understanding a country's current position relative to others and comparing its performance over time provides valuable insights for redefining and optimizing policies. Given the high correlation between VOCs, it is crucial to leverage insights gained from previous VOCs.

Machine learning (ML) techniques, particularly transfer learning, offer a valuable approach in this context. Transfer learning involves utilizing knowledge gained from solving one problem and applying it to a different but related problem. In the case of forecasting VOCs, transfer learning allows for the efficient incorporation of prior knowledge into the predictive models. This approach enhances the accuracy and effectiveness of VOC prediction models by leveraging the patterns and features learned from the analysis of earlier variants (*Xu et al., 2024*).

Moreover, employing multi-attribution decision-making (MADM) techniques allows for a comprehensive assessment of a country's performance, both in comparison with its past and relative to its counterparts in various VOCs. This approach aids in identifying performance changes over time and evaluating the growth trajectory of the country in
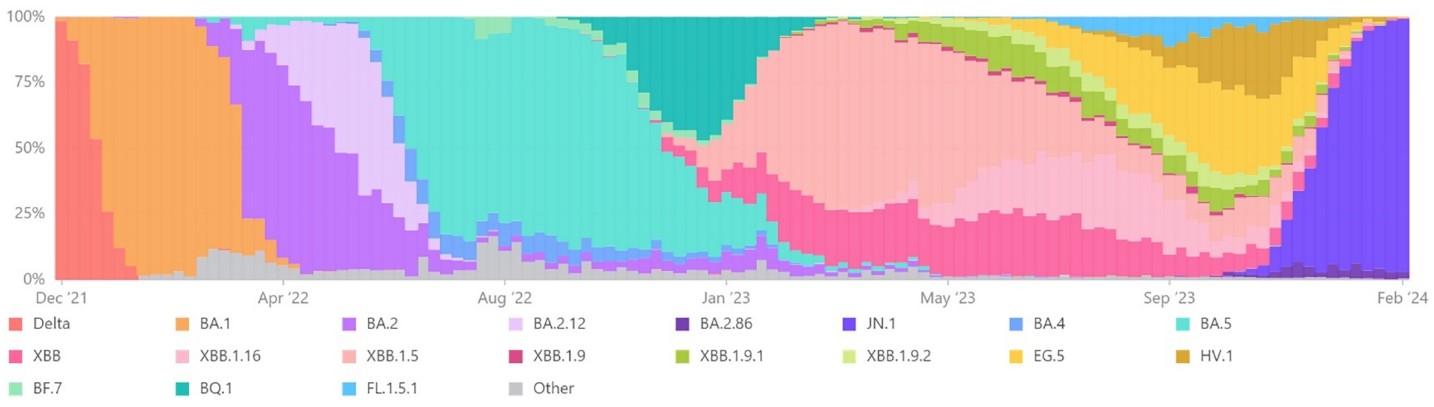

**Figure 1 Nationwide VOCs trends (Dec 2021–Feb 2024).**     

contrast to other nations. While our article is the first to specifically address decision support systems for VOC using integrated MADM and transfer learning techniques, it is worth noting that numerous studies have previously investigated the application of various ML algorithms to forecast COVID-19 spreads and develop decision support systems for the pandemic.

*Fayemiwo et al. (2021)* proposed a Deep transfer learning model (DTL) utilizing fine-tuned VGG-16 and VGG-19 convolutional neural networks (CNNs) for COVID-19 detection from chest X-ray images. The VGG-16 DTL model exhibited superior performance, achieving 99.23% accuracy in binary classification and 93.85% accuracy in three-class classification. *Bahgat et al. (2021)* proposed an Optimized Transfer Learning-based Approach for Automatic Detection of COVID-19 (OTLD-COVID-19), integrating the Manta-Ray Foraging Optimization (MRFO) algorithm to optimize CNN architectures for classifying COVID-19 pneumonia from other types. The study utilized chest X-ray images from various public datasets and found DenseNet121 to yield the highest performance, achieving 98.47% accuracy and robust evaluation metrics. *Saeed et al. (2022)* proposed a novel mathematical framework, the complex fuzzy hypersoft ($\mathcal{CFHS}$) set, integrating complex fuzzy sets and hypersoft sets to improve COVID-19 diagnosis and treatment, effectively handling uncertainty and data complexity. The framework was validated by linking symptoms to medications using $\mathcal{CFHS}$-mapping. *Cai et al. (2022)* proposed a retrospective study to assess the impact of compliance with a respiratory decision support system on COVID-19-associated ARDS patients requiring invasive mechanical ventilation. The results demonstrated a significant association between higher respiratory support decision scores and improved 28-day survival, indicating the potential benefits of the decision support system in managing these patients. *Shen et al. (2022)* proposed a Bayesian networks-based decision support model for COVID-19 risk assessment, utilizing expert knowledge and ML. The model's reliability was demonstrated through comparison with other ML models, suggesting its potential as a global tool for accurate severity assessment and epidemic risk management by healthcare professionals. *Karakosta et al. (2021)* offered a study addressing the energy sector crisis amid the

COVID-19 pandemic, emphasizing the need for targeted energy efficiency investments. The research highlighted the lack of decision support tools for identifying sustainable investments, recommending the Triple-A Horizon 2020 standard as crucial for assessing and financing energy efficiency projects.

*Mete et al. (2023)* suggested an innovative two-phase approach combining the Multi-Choice Best-Worst Method (BWM) and Complex Proportional Assessment of Alternatives (COPRAS) methods to reassess COVID-19 risks in 29 countries, using indicators from the INFORM COVID-19 risk index. This approach provided valuable insights for effective planning and response efforts. *Ismail et al. (2024)* conducted a study on the impact of the COVID-19 pandemic on environmental governance decisions in publicly listed European companies. The research revealed a significantly positive influence of COVID-19 on environmental governance, emphasizing the challenges companies face in maintaining sustainability efforts during the crisis. *Abdulkareem et al. (2022)* suggested a multidimensional examination framework (MEF) for prioritizing COVID-19 severe patients using combined MADM methods. The framework employed the CRITIC method to identify objective weights and the Vise Kriterijumska Optimizacija I Kompromisno Resenje (VIKOR) method to prioritize patients, highlighting the importance of heart disease, cough, and nasal congestion in prioritization. *Arshad et al. (2024)* applied an MADM algorithm using interval-valued multi-fuzzy hypersoft sets to optimize antivirus mask selection during COVID-19, considering several criteria such as effectiveness, comfort, and cost. This method integrated expert opinions and empirical data to handle uncertainty, enhancing decision accuracy and reliability. *Alsattar et al. (2024)* proposed a novel Dynamic Localisation-Based Decision (DLBD) method with Fuzzy Weighting With Zero Inconsistency (FWZIC) in a Probabilistic Single-Valued Neutrosophic Hesitant Fuzzy Set (PSVNHFS) environment for benchmarking Hybrid Multi Deep Transfer and Machine Learning (HMDTML) models, effectively addressing vagueness and uncertainty in COVID-19 chest X-ray images. Results showed Model M24 ranked highest, confirming the proposed method's reliability and robustness. *Abdullah, Kedir & Takore (2024)* presented a hybrid deep learning CNN model with transfer learning, utilizing pre-trained structures like VGG16 and VGG19, achieving 92% accuracy in COVID-19 diagnosis with SVM-linear and neural networks, aiding in accurate diagnosis and validation of positive cases. *Jeon et al. (2023)* proposed a hybrid Fuzzy Multi-Criteria Decision-Making (F-MCDM) model for assessing government strategies during the COVID-19 pandemic, conducting an empirical case study in India with criteria such as acceptance, effectiveness, cost, and simplicity. The top-ranked strategies were vaccinations, social isolation, and emergency development, with implications discussed for developing nations. *Aydin & Yurdakul (2020)* suggested a three-staged framework using Data Envelopment Analysis (DEA) and ML to assess the COVID-19 performances of 142 countries, revealing optimal clustering at three groups and identifying influential parameters such as GDP, smoking rates, and diabetes rates, while highlighting that these factors do not significantly impact countries' effectiveness levels.

Although the existing literature provides various decision support systems to address the challenges posed by COVID-19, notable gaps remain within this field. In this regard:

I) Insufficient attention has been given to the separate consideration of data and decision support systems specifically tailored to VOC. A thorough investigation into each variant can yield valuable insights, enabling the development of more effective policies to address future VOC.

II) There is a deficiency in the assessment of each country's performance in managing different VOCs, and a comparative analysis with other nations is lacking. Evaluating the effectiveness of policies across countries for each VOC can offer valuable benchmarks and facilitate informed decision-making.

III) There is a notable gap in leveraging prior results related to VOCs for future forecasting. By analyzing similarities between different VOCs, the utilization of past VOCs' outcomes can be instrumental in enhancing the accuracy of forecasting the spread of future VOCs. This approach can contribute to a more comprehensive and proactive strategy in managing the ongoing challenges posed by the evolving landscape of COVID-19.

This research addresses the identified gaps in the existing literature by presenting the following key contributions:

- A Variant-Informed Decision Support System (VIDSS) is proposed and designed to adapt dynamically to each VOC. By tailoring the decision support system to the unique characteristics of each VOC, VIDSS aims to establish more effective policies, addressing the critical need for variant-specific strategies.
- The article employs MADM techniques to introduce a novel criterion. This criterion evaluates a country's performance by considering its improvement relative to its past state and concurrently compares it with the improvements of other countries. This approach provides a comprehensive assessment, allowing for nuanced policy comparisons among nations facing distinct VOC challenges.
- Leveraging transfer learning, the study utilizes knowledge from forecast models of past VOCs to enhance predictions for future VOCs. By integrating insights gained from prior VOCs, this method contributes to more accurate and informed forecasting, filling the gap in utilizing historical data for proactive decision-making in the face of evolving challenges posed by COVID-19.

The existing literature reveals several gaps in decision support systems for COVID-19, particularly in the separate consideration of data and systems tailored to VOCs. To address this, the article introduces a VIDSS that dynamically adapts to each VOC, providing tailored strategies. Additionally, the lack of comparative performance assessments among countries and the underutilization of prior results for future forecasting is addressed through the use of MADM techniques, which create a novel criterion for evaluating and comparing countries' performances in managing VOCs. Furthermore, the notable gap in leveraging prior results related to VOC for future forecasting is filled by utilizing transfer learning. This approach analyzes similarities between different VOCs and integrates insights gained from past VOC to enhance predictions for future VOCs, thereby

offering a comprehensive and proactive approach to managing the evolving COVID-19 landscape.

The rest of this article is organized as follows: "Framework and Preliminaries" provides a detailed explanation of the proposed framework and its preliminaries. In "Results", the findings and computational results of the study are presented. "Managerial Insights" offers a comprehensive discussion and provide managerial insights. Finally, "Conclusion and Future Research Directions" provides final remarks, discuss limitations, and outline potential avenues for future research.

# FRAMEWORK AND PRELIMINARIES

In this section, we present a thorough overview of the methodology employed in our study, commencing with essential preliminaries crucial for constructing the framework. This encompasses a comprehensive understanding of the dataset, MADM approaches, and transfer learning. Subsequently, we delve into a meticulous presentation of the VIDSS, providing an in-depth exploration of its intricacies and functionalities.

## Preliminaries

In this section, we provide essential preliminaries by offering detailed insights into the dataset description and MADM methods. Subsequently, we provide a succinct yet informative overview of transfer learning, laying the groundwork for a comprehensive understanding of the framework employed in our study.

### Dataset description

The basis of our analysis in this study lies in the COVID-19 world dataset (https://github.com/owid/covid-19-data/tree/master/public/data) sourced from Our World in Data (https://ourworldindata.org/coronavirus) (*Our World In Data, 2024*). This extensive dataset spans the entire duration of the COVID-19 pandemic, with daily updates meticulously recorded up to March 5, 2024 (*Hasell et al., 2020*; *Mathieu et al., 2021*). In this study, the investigation revolves around the analysis of VOCs, with a particular focus on the integration of COVID-19 variants data sourced from ECDC (https://www.ecdc.europa.eu/en/publications-data/data-virus-variants-covid-19-eueea) and GISAID (https://www.gisaid.org) (*ECDC, 2023*; *GISAID, 2023*). This comprehensive dataset (https://opendata.ecdc.europa.eu/covid19/virusvariant/) encompasses the incidence of cases within each country, categorized by specific VOCs and subvariants. Before exploring the details of the dataset features, it is imperative to delineate the preprocessing procedures undertaken for both features and observations:

1) In the COVID-19 variant dataset, all subvariants are relabeled with their corresponding major VOCs. For instance, subvariants like BA.1 are renamed to their major VOCs, such as Omicron.

2) The COVID-19 variant dataset is streamlined to focus solely on the investigation of the five major VOC: Alpha, Beta, Gamma, Delta, and Omicron.

3) Features in the COVID-19 world dataset are aggregated based on the year-week timeframe, aligning with the temporal resolution in the COVID-19 variant dataset.

4) The COVID-19 world dataset is filtered to exclusively include countries present in the COVID-19 variants dataset.

5) Both datasets are merged based on country and year-week features, fostering a comprehensive alignment of information.

6) The merged dataset is further aggregated based on country and VOCs.

7) Null columns are eliminated, and any missing values are imputed with the median, chosen for its robustness in handling potential outliers.

After the execution of these preprocessing steps, the characteristics of the integrated dataset features are elucidated in Table 1.

### MADM techniques

Utilizing MADM techniques necessitates the application of both weighting and ranking methods. In this context, we employ the Criteria Importance Through the Intercriteria Correlation (CRITIC) method for weighting the criteria and the Combined Compromise Solution (CoCoSo) technique to rank the alternatives. We utilize the well-established CRITIC method for weighting, acknowledged for its robustness in decision-making contexts. Additionally, we employ the CoCoSo method, a novel approach in decision science, to rank alternatives. Notably, our focus lies on deriving pivotal policies through country comparisons, rather than a comparative analysis of decision-making methodologies.

The CRITIC technique is used to assign weights to attributes based on a decision matrix, ensuring coherence without contradictions. Notably, CRITIC offers advantages such as accounting for interdependencies between attributes and eliminating the need for attribute independence. Moreover, qualitative attributes can be effectively quantified using this approach. The CRITIC method follows a systematic four-stage process, utilizing correlation coefficients to identify attribute relationships, transforming qualitative attributes into quantitative measures, and ultimately determining the superior attribute. For detailed steps on the CRITIC method, readers can refer to the provided references (*Diakoulaki, Mavrotas & Papayannakis, 1995*; *Silva et al., 2023*).

On the other hand, the CoCoSo ranking method integrates a hybrid model, combining simple additive weighting (SAW) and exponentially weighted product (EWP) approaches. It serves as a versatile tool for generating compromise solutions in decision-making scenarios. To address a CoCoSo decision problem, one must identify alternatives and relevant criteria. The subsequent validated steps involve normalization, weight assignment, SAW calculation, EWP application, and the derivation of a comprehensive compromise solution. Readers can refer to the provided references for a thorough explanation of the CoCoSo method steps (*Yazdani et al., 2019*; *Pajić, Andrejić & Kilibarda, 2022*).

### Transfer learning

Transfer learning is a prevalent ML approach that involves leveraging a model previously trained on a specific task and applying it to a new, related task. This technique is particularly popular in the realm of deep learning, as it enables the training of models

**Table 1 Description of integrated COVID-19 dataset's features.**

| Features | Description |
|---|---|
| Aged 65-older | Percentage of the population aged 65 and older, based on the most recent available data |
| Aged 70-older | Percentage of the population aged 70 and older in the year 2015 |
| Cardiovascular death rate | Annual mortality rate of cardiovascular disease, expressed as deaths per 100,000 individuals |
| Country | Refers to the name of a nation |
| Diabetes prevalence | Percentage of individuals between 20 to 79 years old within a specific population who have been diagnosed with diabetes |
| Excess mortality | Percentage disparity between reported weekly or monthly deaths in 2020–2021 and anticipated deaths based on previous years |
| Extreme poverty | Percentage of individuals living in conditions of extreme poverty within a specific population |
| Female smokers | Percentage of women who smoke cigarettes |
| Gross Domestic Product (GDP) Per Capita | Gross domestic product at purchasing power parity, presented in constant 2011 international dollars (most recent year available) |
| Hospital patients | Count of COVID-19 patients in hospitals |
| Hospital.Beds.Per.100K | Number of hospital beds available for every 1,000 individuals within a specific population |
| HDI | A composite indicator assessing the average level of success in health, education, and standard of living |
| ICU patients | Count of COVID-19 patients in Intensive Care Units (ICUs) |
| Life expectancy | Average lifespan at birth in the year 2019 |
| Male smokers | Percentage of men who smoke cigarettes |
| Median age | Median age of the population based on UN projections for 2020 |
| People fully vaccinated | Total number of individuals who received all doses prescribed by the initial vaccination protocol |
| People vaccinated | Total number of individuals who received at least one vaccine dose |
| Population | Refers to the total population size of the country |
| Population density | Number of people per square kilometer of land area, based on the most recent available data |
| Positive rate | Proportion of COVID-19 tests returning positive results, presented as a rolling 7-day average |
| Reproduction rate | Real-time estimate of the effective reproduction rate (R) of COVID-19 |
| Stringency index | A composite measure indicating the strictness of government responses to COVID-19, derived from nine indicators (100 = strictest response) |
| Total cases | Total confirmed cases of COVID-19 in each VOC |
| Total tests | Cumulative number of tests conducted for COVID-19 |
| Total vaccination | Overall count of COVID-19 vaccination doses administered |
| Variant | Refers to the specific COVID-19 variant |
| Weekly hospital admissions | Count of newly admitted COVID-19 patients to hospitals |
| Weekly ICU admissions | Count of newly admitted COVID-19 patients to intensive care units (ICUs) |

based on deep neural networks even when confronted with limited datasets (*Ibrahim & Tapamo, 2024*). Employing transfer learning proves beneficial in mitigating the challenges associated with training, especially when dealing with a small training set (*Fu, Zhu & Li, 2019*). Transfer learning encompasses four distinct types: instance-based transfer, feature-based transfer, parameter-based transfer, and relational-based transfer, offering a foundational framework for understanding various approaches in transfer learning and serving as a platform for the development of innovative methods (*Pan & Yang, 2009*). Instance-based transfer involves transferring knowledge at the individual instance level,

allowing models to leverage prior experience from one task to another. Feature-based transfer entails transferring knowledge at the feature level, where relevant attributes learned from one task inform a related task. Parameter-based transfer involves sharing model parameters between tasks, enabling adaptation of learned parameters from one task to another. Relational-based transfer focuses on transferring knowledge about the relationships between data entities, aiding the model's comprehension of connections in a new task based on prior task experiences (*Toscano-Miranda et al., 2024*).

## Proposed VIDSS

To enhance the clarity and coherence of the proposed VIDSS, we present a detailed explanation in Fig. 2. The VIDSS process initiates with the acquisition of a comprehensive dataset, integrating information on the spread of COVID-19 associated with each VOC and the relevant features of countries affected. This dataset is further categorized, with each subset filtered by individual VOCs in Stages (1) and (2). To facilitate understanding, we investigate the relationships between two VOC datasets, establishing a foundation for subsequent analyses.

For VOC 1, Stage (3) involves weighting each feature using the CRITIC method. In Stages (4) and (5), the CoCoSo method is applied to assess countries' performance based on these weights, assigning scores accordingly. The only parameter setting of CoCoSo is the λ parameter, which is set to 0.5 to balance the importance of both the subjective and objective components in the decision-making process. In Stage (6), parameters for the neural network are selected, initially chosen randomly for the first model. Subsequently, Stage (7) involves training the model, while Stage (9) unveils feature importance, elucidating the influential factors in the spread of the virus. In Stage (8), post-training, the model's parameters, including weights and biases, are saved for future VOC models.

For VOC 2, analogous actions are undertaken in Stages (10) to (12) to calculate scores for each country. The novel criterion, the Relative Performance Index (RPI), is introduced in Stage (13), considering a country's performance relative to its performance in prior VOCs and its growth compared to other countries. Equations (1) and (2) provides the RPI formula:

$$G_i = \frac{S_{ij} - S_{i(j-1)}}{S_{i(j-1)}}, \tag{1}$$

$$RPI_i = G_i - \frac{1}{n}\sum_{i=1}^{} G_i, \tag{2}$$

where $G_i$ represents the growth of country $i$, and $S_{ij}$ signifies the score of country $i$ in VOCs $j$.

Transfer learning is applied in Stage (14), utilizing parameters trained in the prior VOC model. The subsequent training of the model occurs in Stages (16) and (17), determining the most crucial features. At this juncture, policymakers can derive essential insights to refine their strategies. Stage (18) highlights the first tool: feature importance, which may differ across VOCs. Identifying the most influential features in each VOC pandemic aids in policy formulation. Stage (19) introduces the second tool for policymakers–RPI,

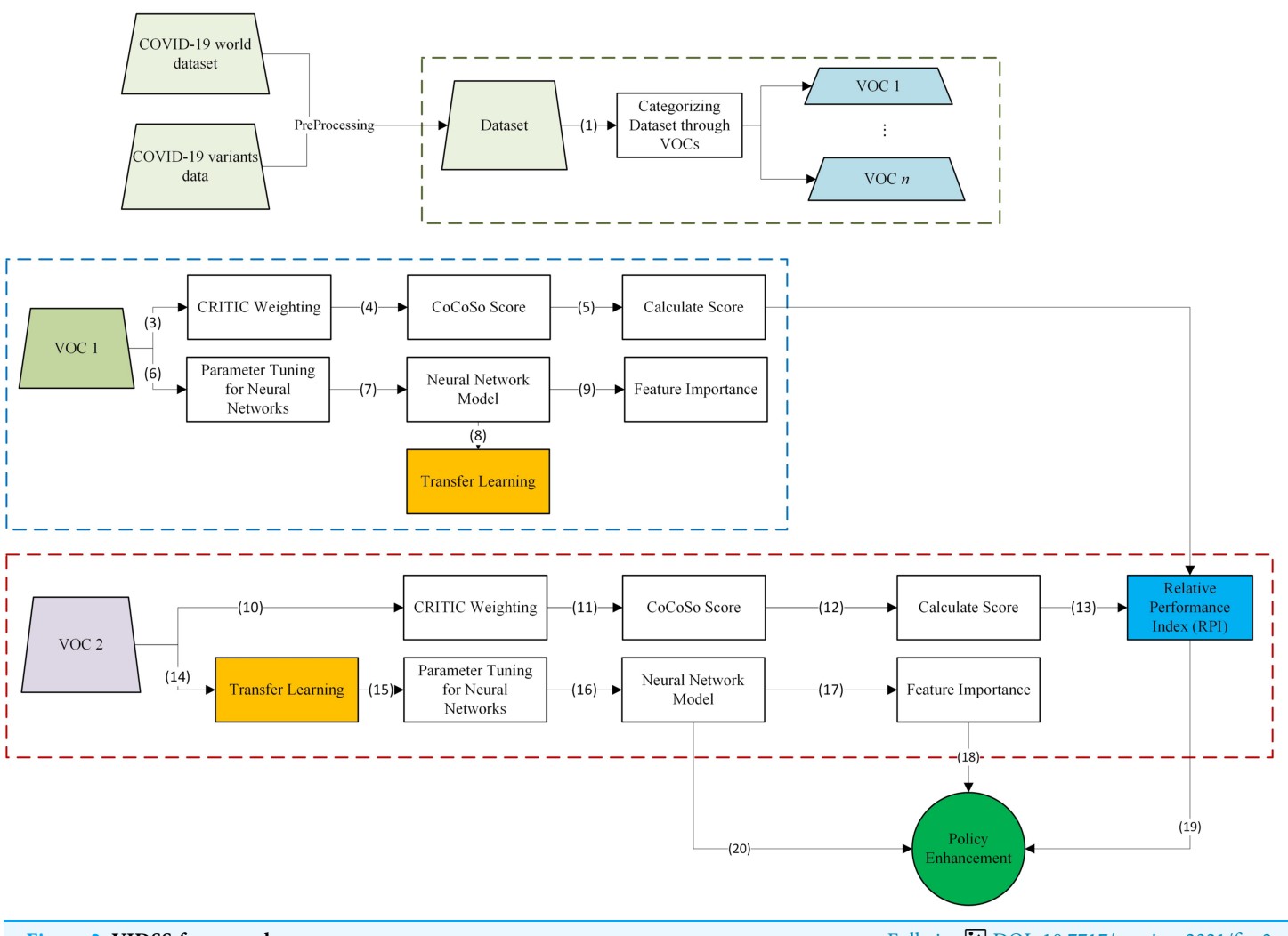

**Figure 2 VIDSS framework.**

illustrating a country's performance from different perspectives. Lastly, Stage (20) introduces the model of spread, allowing efficient forecasting to assist countries in strategic planning.

In summary, our proposed VIDSS method offers a systematic approach to analyzing COVID-19 data by focusing on each VOC individually. The process starts with the collection and categorization of data for each VOC. Using the CRITIC method, we assign weights to various features, and the CoCoSo method evaluates countries' performances based on these weighted features. By incorporating transfer learning, we leverage knowledge from previous VOCs to inform initial weights for future models, thereby enhancing prediction accuracy. Key tools, such as feature importance and the newly introduced RPI, provide policymakers with critical insights into the most influential factors in virus spread and a comparative assessment of countries' performances. This

comprehensive approach aims to improve decision-making and strategic planning in effectively managing the COVID-19 pandemic.

## RESULTS

In this section, the outcomes of the VIDSS framework are meticulously examined, with a structured discussion on the various VOCs in a specific order based on their chronological emergence, namely Alpha, Beta, Gamma, Delta, and Omicron. Commencing with the Alpha variant, the initial dataset undergoes processing within the framework. The neural network model is initialized with random parameters, and after the training phase, the learned weights and biases are stored for use in the subsequent variant. The study employs a rigorous K-fold cross-validation approach with $k = 10$ to robustly assess model performance across multiple folds. This method ensures comprehensive evaluation of predictive accuracy while mitigating bias from data partitioning, enhancing the study's reliability and generalizability. Feature importance is vividly portrayed through SHAP plots, shedding light on the significance of various features in the model's decision-making process. Simultaneously, the CRITIC-CoCoSo approach integrated approaches yield scores for each country, offering valuable insights into their performance. These scores are generated based on positive and negative criteria, where higher scores indicate better performance in terms of spread for positive criteria, and conversely, lower scores denote efficiency in curbing spread for negative criteria. Consequently, a lower value signifies a superior performance index for each country.

Table 2 categorizes criteria into positive and negative factors influencing COVID-19 spread. Positive criteria, such as high population density, median age, and positive rate, are indicative of conditions that may contribute to the rapid spread of COVID-19. These factors suggest increased human interaction, an older population susceptible to severe outcomes, and elevated infection rates, respectively, fostering an environment conducive to transmission. Conversely, negative criteria, including high GDP *per capita*, HDI, and life expectancy, reflect socio-economic and health indicators associated with lower COVID-19 spread. Countries with stronger economies, higher human development, and longer life expectancy often possess better healthcare infrastructures, healthcare practices, and societal well-being, contributing to more effective containment and lower transmission rates.

Moving to the Beta variant, the concept of transfer learning is applied, utilizing the weights and biases derived from the Alpha model as initial parameters for the neural network. Feature importance is represented through SHAP plots. After scoring *via* MADM methods, the RPI criterion is derived. Policymakers can strategically optimize their approaches by integrating insights from the RPI, feature importance, and the forecast model. This coherent approach is consistently applied across subsequent variants, maintaining the interconnected relationships between datasets. Comprehensive neural network parameter settings are succinctly outlined in Table 3, ensuring transparency and reproducibility.

According to Table 3, The chosen hyperparameters for our neural network model were carefully selected to balance performance and generalization. We set the learning rate to

**Table 2 MADM positive and negative criteria.**

| Positive criteria | Negative criteria |
|---|---|
| Aged_65_older | Gdp_per_capita |
| Aged_70_older | Human_development_index |
| Diabetes_prevalence | Life_expectancy |
| Excess_mortality | New_tests |
| Extreme_poverty | People_fully_vaccinated |
| Female_smokers | People_vaccinated |
| Median_age | Stringency_index |
| Population_density | Total_vaccinations |
| Positive_rate | |
| Weekly_hosp_admissions | |
| Population | |
| Weekly_icu_admissions | |
| Cardiovasc_death_rate | |
| Number_detections_variant | |
| Hosp_patients | |
| Male_smokers | |
| ICU_patients | |
| Reproduction_rate | |

**Table 3 Parameter setting for the neural network model.**

| Parameter | Value |
|---|---|
| Learning rate | 0.001 |
| Epochs | 300 |
| Hidden layers | 5 |
| Neurons per layer | 128, 64, 32, 16, 1 |
| Activation function | ReLU |
| Dropout rate | 0.6 (first two layers), 0.3 (remaining layers) |
| Regularizers | L2 (0.0001) |
| Optimizer | Adam |
| Loss function | Mean Absolute Error (MAE) |
| Output classes | 1 |
| Validation split | 0.3 |
| Early stopping | Yes (patience = 150) |

0.001 to ensure stable and efficient optimization using the Adam optimizer, known for its adaptive learning rate capabilities. The model was trained for 300 epochs, with early stopping patience set to 150 epochs, allowing the model to halt training when validation performance ceased to improve, thus preventing overfitting. Our architecture consisted of five hidden layers with a decreasing number of neurons (128, 64, 32, 16, and 1), using the

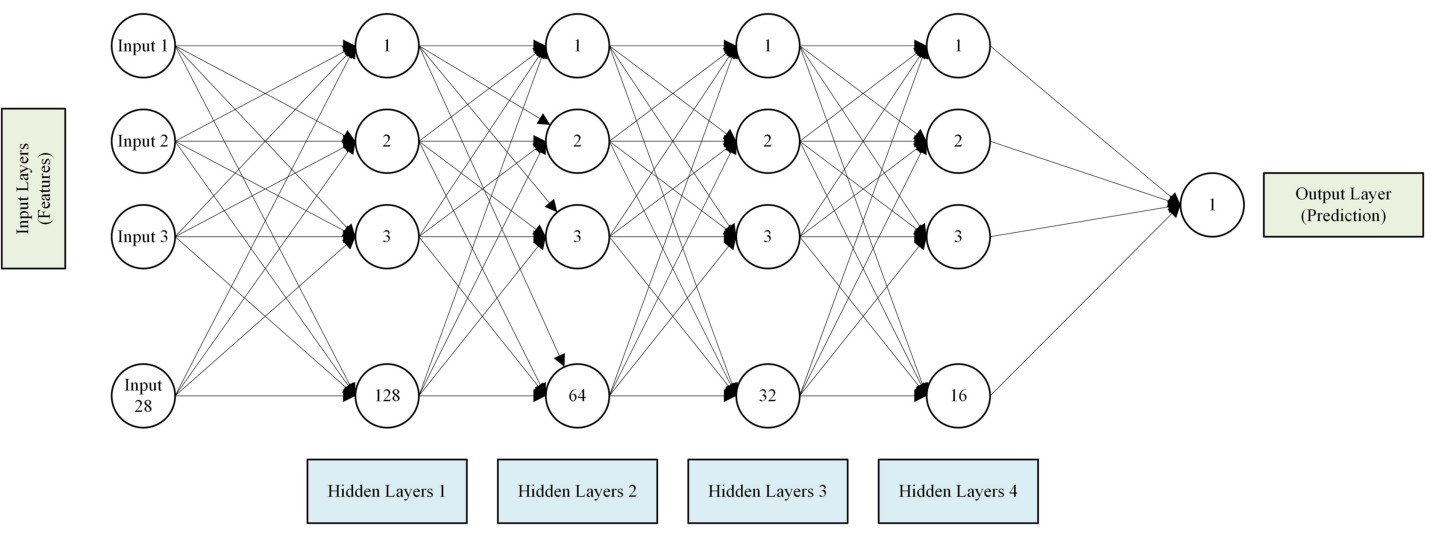

**Figure 3 Architecture of the neural network model.**

ReLU activation function to introduce non-linearity and improve the learning of complex patterns. Dropout rates were set at 0.6 for the first two layers and 0.3 for the subsequent layers to reduce overfitting by randomly deactivating neurons during training. L2 regularization with a coefficient of 0.0001 was applied to further prevent overfitting by penalizing large weights. The Mean Absolute Error (MAE) was chosen as the loss function due to its robustness in handling outliers. A validation split of 0.3 was used to allocate sufficient data for validating the model's performance during training. These settings were designed to create a balanced model capable of generalizing well to unseen data.

Figure 3 illustrates the architecture of our neural network model, comprising an input layer, five hidden layers with progressively fewer neurons, and an output layer. The ReLU activation function was selected for the hidden layers due to its computational efficiency and ability to mitigate the vanishing gradient problem, which is common with other activation functions like sigmoid and tanh. ReLU enhances the model's ability to learn complex patterns and converges faster during training. Adjustments to each parameter in the neural network can significantly impact the model's performance and generalization ability. The learning rate influences the speed and stability of optimization; altering it could lead to faster convergence or instability. The number of epochs and early stopping patience affect how long the model trains, potentially affecting underfitting or overfitting. Changes in the number of neurons per layer impact the model's capacity to learn complex features, while adjustments to activation functions like ReLU alter how nonlinear relationships are captured. Dropout rates and regularization coefficients manage overfitting; modifying these can either enhance generalization or decrease model performance. The choice of loss function and validation split influences how well the model handles data characteristics and assesses its performance. Each parameter's tuning is empirical, impacting the model's ability to generalize to unseen data, highlighting the delicate balance required in neural network configuration.

**Table 4 Comparison of the MADM scores, growth percentage, and RPI percentage among VOCs in different countries.**

| Country | Score-alpha | Score-beta | G | RPI | Score-gamma | G | RPI | Score-delta | G | RPI | Score-omicron | G | RPI |
|---|---|---|---|---|---|---|---|---|---|---|---|---|---|
| Austria | 1.605 | 1.601 | −0.235% | 0.946% | 1.611 | 0.575% | 0.291% | 1.601 | −0.596% | −0.753% | 1.579 | −1.396% | −0.513% |
| Belgium | 1.672 | 1.650 | −1.326% | −0.113% | 1.652 | 0.147% | −0.127% | 1.648 | −0.292% | −0.475% | 1.613 | −2.085% | −1.220% |
| Bulgaria | 1.896 | 1.873 | −1.215% | −0.006% | 1.874 | 0.093% | −0.185% | 1.877 | 0.136% | −0.064% | 1.835 | −2.241% | −1.420% |
| Croatia | 1.834 | 1.818 | −0.916% | 0.293% | 1.820 | 0.123% | −0.162% | 1.826 | 0.341% | 0.139% | 1.783 | −2.376% | −1.608% |
| Cyprus | 1.431 | 1.434 | 0.205% | 1.425% | 1.423 | −0.781% | −1.072% | 1.431 | 0.596% | 0.399% | 1.438 | 0.512% | 1.219% |
| Czechia | 1.791 | 1.768 | −1.249% | 0.028% | 1.771 | 0.180% | −0.154% | 1.753 | −1.028% | −1.209% | 1.765 | 0.671% | 1.427% |
| Denmark | 1.582 | 1.563 | −1.205% | 0.074% | 1.581 | 1.205% | 0.864% | 1.594 | 0.782% | 0.551% | 1.562 | −2.014% | −1.199% |
| Estonia | 1.744 | 1.723 | −1.177% | 0.105% | 1.729 | 0.350% | 0.047% | 1.733 | 0.201% | −0.007% | 1.737 | 0.230% | 0.993% |
| Finland | 1.645 | 1.623 | −1.306% | −0.020% | 1.636 | 0.812% | 0.511% | 1.637 | 0.042% | −0.166% | 1.605 | −1.922% | −1.114% |
| France | 1.701 | 1.684 | −1.020% | 0.265% | 1.687 | 0.152% | −0.125% | 1.687 | 0.033% | −0.183% | 1.681 | −0.388% | 0.367% |
| Germany | 1.687 | 1.678 | −0.514% | 0.785% | 1.683 | 0.255% | −0.028% | 1.678 | −0.278% | −0.502% | 1.670 | −0.446% | 0.328% |
| Greece | 1.709 | 1.652 | −3.349% | −2.009% | 1.699 | 2.875% | 2.591% | 1.654 | −2.656% | −2.907% | 1.650 | −0.211% | 0.580% |
| Hungary | 1.762 | 1.741 | −1.218% | 0.011% | 1.740 | −0.053% | −0.193% | 1.754 | 0.844% | 0.431% | 1.768 | 0.801% | 1.624% |
| Iceland | 1.270 | 1.290 | 1.536% | 2.765% | 1.277 | −0.948% | −1.100% | 1.268 | −0.705% | −1.093% | 1.264 | −0.342% | 0.577% |
| Ireland | 1.434 | 1.413 | −1.465% | −0.063% | 1.422 | 0.641% | 0.421% | 1.429 | 0.477% | 0.021% | 1.439 | 0.706% | 1.660% |
| Italy | 1.694 | 1.699 | 0.274% | 1.672% | 1.662 | −2.191% | −2.383% | 1.684 | 1.357% | 0.903% | 1.680 | −0.250% | 0.816% |
| Latvia | 1.819 | 1.797 | −1.222% | 0.295% | 1.807 | 0.557% | 0.194% | 1.806 | −0.030% | −0.420% | 1.809 | 0.149% | 1.272% |
| Liechtenstein | 1.549 | 1.529 | −1.304% | 0.235% | 1.529 | −0.001% | −0.349% | 1.530 | 0.055% | −0.367% | 1.490 | −2.590% | −1.369% |
| Lithuania | 1.783 | 1.763 | −1.088% | 0.471% | 1.764 | 0.020% | −0.357% | 1.771 | 0.387% | −0.066% | 1.764 | −0.360% | 0.747% |
| Luxembourg | 1.423 | 1.402 | −1.522% | 0.080% | 1.410 | 0.614% | 0.205% | 1.418 | 0.546% | 0.087% | 1.402 | −1.116% | 0.059% |
| Malta | 1.622 | 1.587 | −2.171% | −0.560% | 1.607 | 1.268% | 0.879% | 1.599 | −0.521% | −0.971% | 1.612 | 0.832% | 2.013% |
| Netherlands | 1.682 | 1.660 | −1.321% | 0.227% | 1.671 | 0.666% | 0.375% | 1.672 | 0.061% | −0.497% | 1.677 | 0.276% | 1.681% |
| Norway | 1.487 | 1.469 | −1.222% | 0.354% | 1.481 | 0.810% | 0.566% | 1.488 | 0.491% | −0.129% | 1.495 | 0.441% | 2.056% |
| Poland | 1.698 | 1.678 | −1.200% | 0.427% | 1.677 | −0.022% | −0.185% | 1.683 | 0.333% | −0.306% | 1.638 | −2.680% | −0.771% |
| Portugal | 1.725 | 1.704 | −1.195% | 0.504% | 1.707 | 0.164% | −0.030% | 1.703 | −0.197% | −0.887% | 1.672 | −1.833% | −0.053% |
| Romania | 1.835 | 1.770 | −3.575% | −1.776% | 1.773 | 0.209% | 0.009% | 1.820 | 2.653% | 1.786% | 1.732 | −4.846% | −3.077% |
| Slovakia | 1.719 | 1.693 | −1.515% | −0.160% | 1.706 | 0.729% | 0.531% | 1.712 | 0.356% | −0.065% | 1.709 | −0.172% | 0.828% |
| Slovenia | 1.708 | 1.688 | −1.222% | 0.079% | 1.692 | 0.258% | 0.237% | 1.696 | 0.251% | −0.191% | 1.693 | −0.189% | 1.087% |
| Spain | 1.778 | 1.748 | −1.689% | −0.348% | 1.735 | −0.712% | −0.615% | 1.746 | 0.606% | 0.069% | 1.717 | −1.632% | 0.188% |
| Sweden | 1.619 | 1.602 | −0.993% | 0.000% | 1.611 | 0.517% | 0.000% | 1.618 | 0.469% | 0.000% | 1.586 | −2.008% | 0.000% |

In Table 4, the comparison of MADM scores, growth percentages, and RPI percentages among VOCs in different countries is presented. It is crucial to interpret these scores contextually, recognizing that individual country scores gain significance when assessed against past performance and other countries. Take Austria, for instance; despite a −0.235% growth in the Beta variant compared to the Alpha variant, the 0.946% RPI signifies a commendable performance, considering the intricate nature of each variant. Similarly, Belgium's 0.147% growth rate in the Gamma variant suggests improvement compared to the Beta variant, yet the −0.127% RPI underscores that the growth pace, relative to other countries, remains insufficient. By integrating both growth and RPI

**Table 5 MAE, MSE and RMSE results for the neural network models predictions comparing performance with and without transfer learning across VOCs.**

| Variant | Metric | With transfer learning | Without transfer learning |
|---------|--------|------------------------|---------------------------|
| Beta | MAE | 2,736,003.20 | 6,315,651.13 |
| | MSE | 28,565,633,048,668.29 | 154,804,758,974,755.94 |
| | RMSE | 3,912,237.48 | 9,368,524.90 |
| Gamma | MAE | 5,135,502.21 | 12,279,910.28 |
| | MSE | 86,162,363,661,281.53 | 574,019,783,996,581.88 |
| | RMSE | 6,981,456.68 | 18,078,069.78 |
| Delta | MAE | 3,992,971.00 | 9,111,414.52 |
| | MSE | 63,241,661,533,989.48 | 325,626,436,466,515.31 |
| | RMSE | 5,654,557.16 | 13,431,780.60 |
| Omicron | MAE | 33,699,864.10 | 85,570,949.59 |
| | MSE | 3,890,865,123,209,482.50 | 28,717,687,005,037,884.00 |
| | RMSE | 46,367,353.26 | 125,918,857.70 |

metrics, a more comprehensive evaluation of a country's performance emerges, offering a nuanced understanding of its standing amid evolving variants.

The MAE results in Table 5 demonstrate a clear performance advantage for neural network models employing transfer learning across various VOCs compared to those without. The performance gains are evident across all VOCs, with significantly lower MAE, Mean Squared Error (MSE) and Root Mean Square Error (RMSE) values observed when transfer learning is applied. This suggests that leveraging knowledge from previous VOC models enhances the predictive capabilities of the neural network. The ordered sequence of VOCs allows for cumulative learning, where each subsequent VOC benefits from the knowledge acquired in modeling the preceding ones. This implies that transfer learning can be a potent strategy for effectively predicting upcoming VOCs, as the model gradually accumulates insights from multiple VOCs, leading to more accurate predictions for the last VOCs in the sequence.

According to Fig. 4, the feature importance in predicting COVID-19 cases varies across different variants, highlighting the evolving nature of the pandemic and the corresponding response measures. In the Alpha variant, "people fully vaccinated," "Intensive Care Unit (ICU) patients," and "weekly hospital admissions" emerge as the top features. The high importance of vaccination status indicates a significant impact on the spread of the Alpha variant, with higher vaccination rates contributing to lower transmission and milder cases. The number of ICU patients reflects case severity and healthcare system burden, providing real-time indicators for managing healthcare capacities. Weekly hospital admissions highlight the influx of severe cases requiring hospitalization, assisting in tracking the variant's spread and healthcare response effectiveness.

For the Beta variant, "weekly hospital admissions," "total vaccinations," and "new tests" are the top features. Weekly hospital admissions remain crucial for understanding healthcare burden and variant severity. Total vaccination efforts are essential in mitigating

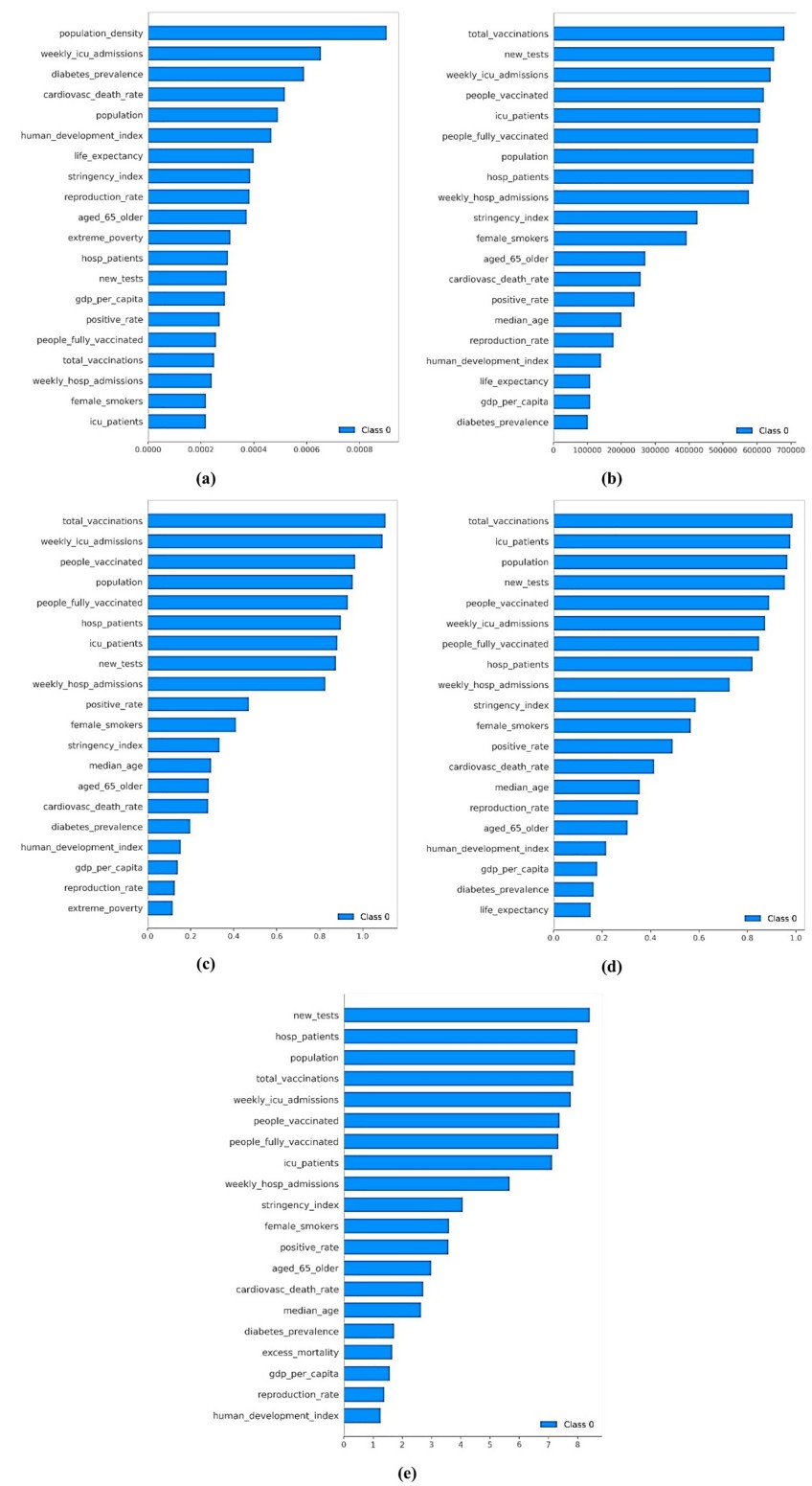

**Figure 4 Mean SHAP values for the feature importance across VOCs: (A) Alpha, (B) Beta, (C) Gamma, (D) Delta, and (E) Omicron.**

the Beta variant's spread and impact, reflecting the population's immunity level. The importance of new tests indicates the need for robust testing infrastructure to identify and isolate cases promptly, controlling the variant's spread.

In the Gamma variant, "weekly hospital admissions," "ICU patients," and "people vaccinated" are most important. Weekly hospital admissions continue to gauge the pandemic's severity. ICU patient numbers remain critical for indicating severe cases and guiding healthcare resource allocation. The role of partial vaccination suggests that even initial doses help control the variant's spread and severity.

For the Delta variant, the top features are "weekly hospital admissions," "ICU patients," and "total vaccinations." Weekly hospital admissions highlight the importance of tracking severe cases. The high number of ICU patients underscores the variant's severity and healthcare strain. Total vaccination efforts are crucial in controlling this highly transmissible variant, reflecting the population's immunity.

In the Omicron variant, "weekly hospital admissions," "total vaccinations," and "people vaccinated" are top features. Despite Omicron being less severe in individual cases, the high number of cases makes hospital admissions crucial. The importance of total vaccinations continues, underscoring immunity's role in managing the spread. The inclusion of partially vaccinated individuals highlights the need for ongoing vaccination campaigns, even for less severe variants.

In conclusion, the analysis of feature importance across different COVID-19 variants reveals critical insights into the pandemic's dynamics and response measures' effectiveness. According to Fig. 3, the consistent importance of weekly hospital admissions and ICU patients across all variants underscores the need for robust healthcare infrastructure to manage severe cases and prevent healthcare system overload. The evolving significance of vaccination-related features highlights the profound impact of vaccination campaigns in mitigating the spread and severity of COVID-19. In earlier variants like Alpha and Beta, the focus on ICU patients and weekly hospital admissions reflects the immediate need to manage severe cases. As the pandemic progresses and vaccination efforts intensify, the importance of total vaccinations and people vaccinated becomes more pronounced, indicating the shifting focus towards achieving herd immunity and reducing overall transmission. The consistent inclusion of testing metrics in the Beta variant emphasizes the critical role of testing infrastructure in promptly identifying and isolating cases, and curbing the spread. These dynamic shifts in feature importance offer valuable insights for shaping and refining pandemic management policies based on evolving scenarios and key determinants. Understanding the varying impact of different features enables policymakers to allocate resources better and implement targeted measures to effectively manage and control the pandemic across different variants.

## MANAGERIAL INSIGHTS

In the aftermath of the COVID-19 pandemic, the global landscape faced unprecedented challenges with the emergence of specific variants, known as VOCs, posing a significant and ongoing threat to global health. The unique genetic changes within these VOCs, influencing virus characteristics and exhibiting heightened transmissibility, triggered

prioritized global scrutiny, extensive research endeavors, and adaptive responses. Governments worldwide found themselves grappling with the unpredictable evolution of VOCs, introducing uncertainties and volatility that hindered their efforts to safeguard their nations.

This work introduces a VIDSS framework aimed at addressing the existing gaps in dynamic decision-making amidst various VOCs. Particularly pertinent in the context of pandemics like COVID-19, where variants and subvariants continuously emerge, these variations often exhibit both differences and similarities. Leveraging insights from prior experiences and VOCs becomes imperative in navigating such complexities. The framework presented herein equips policymakers with indispensable tools, notably forecast models leveraging deep learning techniques enhanced by transfer learning from previous VOCs. These models discern significant features within each VOC, facilitating precise policy formulation. Additionally, the framework incorporates a criterion for evaluating a country's performance relative to its historical trajectory and that of other nations, offering a comprehensive assessment tool. This comprehensive approach enables tailored policy responses to diverse challenges based on past experiences. The forecast models serve multiple purposes, contributing to an early warning system, facilitating resource allocation, optimizing vaccination campaigns, and aiding in economic planning, among other applications. Moreover, the country's performance assessment not only enables benchmarking against peers but also aids in identifying best practices, learning from mistakes, and adapting strategies. Furthermore, the identification of feature importance within each VOC empowers policymakers in making informed decisions, implementing targeted interventions, and evaluating policy effectiveness. By amalgamating these components, the framework offers a robust foundation for navigating the complex landscape of pandemics and other dynamic scenarios, fostering resilience and adaptability in policymaking.

While the VIDSS framework offers a systematic and innovative approach to understanding and managing the spread of VOCs, there are notable limitations to consider. Firstly, the dataset's reliance on reported cases and deaths may be influenced by varying testing capabilities and reporting accuracy across different countries, potentially impacting the model's input quality. Secondly, the focus on major VOCs may overlook the impact of other emerging subvariants, which could also play significant roles in the pandemic's progression. Lastly, the transfer learning approach, while effective, assumes that patterns from previous VOCs are applicable to future ones, which may not always hold true given the virus's potential for significant genetic changes. Addressing these limitations in future research could further refine and enhance the VIDSS framework, providing even more accurate and actionable insights for managing COVID-19 and other similar pandemics.

## CONCLUSION AND FUTURE RESEARCH DIRECTIONS

The COVID-19 pandemic has presented an unprecedented global challenge, impacting various facets of society, the economy, and the environment. The emergence of specific variants designated as VOCs in 2020 intensified the global health threat, prompting the

WHO to prioritize monitoring, research, and adaptive responses. The ever-evolving nature of these VOCs created uncertainties for governments worldwide, hindering their efforts to safeguard their nations effectively. In response to these challenges, this research proposed the implementation of a VIDSS framework, which dynamically adapts to the unique characteristics of each VOC.

This study makes significant theoretical contributions by integrating MADM techniques with transfer learning to address the complex and dynamic nature of COVID-19 VOCs. The introduction of the RPI as a novel criterion provides a comprehensive assessment of a country's performance in managing VOCs, considering both historical and comparative data. This research enhances the understanding of how different factors influence the spread and impact of VOCs, highlighting the importance of adaptable and data-driven approaches in pandemic management.

The VIDSS framework contributes to the field by offering a systematic approach to evaluating and responding to VOCs. It incorporates MADM techniques and transfer learning, enabling the model to adapt and improve predictions based on historical data. The proposed VIDSS framework consists of two main stages: analyzing and comparing countries' performance, and predicting the spread of VOCs. The first stage utilizes MADM tools, specifically CRITIC and CoCoSo methods, to define the RPI. This stage allows for a comprehensive comparison of countries, identifying benchmarks and indicators of performance. This comparative analysis is crucial for understanding each country's strengths and weaknesses in managing VOCs, providing a valuable tool for policymakers to optimize their strategies. The second stage involves predicting the spread of VOCs using neural networks and transfer learning. By leveraging insights from past VOCs, this stage improves the accuracy of forecasts for future variants. This predictive capability is essential for proactive decision-making, enabling countries to prepare and respond effectively to new threats. Additionally, the feature analysis provides valuable insights into the factors influencing the pandemic's progression, emphasizing the importance of vaccination rates, healthcare infrastructure, and socio-economic conditions.

The most crucial insight from the VIDSS framework results is the consistently high importance of vaccination rates and healthcare infrastructure indicators, such as weekly hospital admissions and ICU patients, across all COVID-19 variants. This underscores the significant role of robust vaccination campaigns and strong healthcare systems in mitigating the spread and severity of the virus. For policymakers and countries, this highlights the necessity of investing in comprehensive vaccination efforts and enhancing healthcare capacities to manage severe cases, ensuring a proactive and resilient response to evolving pandemic scenarios. This focus can aid in efficiently allocating resources and implementing targeted measures, ultimately reducing the transmission and impact of COVID-19.

The practical advantages of the VIDSS framework are evident in its ability to provide nuanced policy comparisons among nations facing distinct VOC challenges. By leveraging insights from past VOCs, the framework enhances forecasting accuracy, enabling more informed and proactive decision-making. The VIDSS framework's emphasis on real-time data integration and adaptability ensures that it remains relevant and effective in

addressing the evolving nature of COVID-19 variants. Policymakers can utilize this framework to tailor their strategies to the specific characteristics of each VOC, improving their response to the pandemic.

Despite the promising potential of VIDSS, certain limitations must be acknowledged. The system heavily relies on the availability and accuracy of data related to VOCs, and any limitations or inaccuracies in the data could compromise the system's ability to provide reliable support. The rapid evolution of VOCs poses a challenge in keeping the VIDSS consistently updated and adaptable. Ensuring real-time adjustments to the system to address new variants may be logistically challenging.

Future research should consider integrating additional features that could enhance the reliability and accuracy of the VIDSS framework's forecasting capabilities. This could include more granular health data, socio-economic indicators, and mobility patterns. Additionally, Improvements to the spatial and temporal dimensions of the VIDSS framework could provide a more detailed understanding of the spread dynamics of COVID-19 variants. This could involve the use of geospatial analysis and time-series forecasting techniques. Moreover, developing mechanisms for the real-time integration of emerging data on new variants and subvariants will be crucial for maintaining the VIDSS framework's relevance and effectiveness. This could involve partnerships with health organizations and the use of advanced data-sharing technologies.

### Funding
The authors received no funding for this work.

### Competing Interests
Erfan Babaee Tirkolaee is an Academic Editor for PeerJ.

### Author Contributions
- Amirreza Salehi Amiri conceived and designed the experiments, performed the experiments, analyzed the data, performed the computation work, prepared figures and/or tables, authored or reviewed drafts of the article, and approved the final draft.
- Ardavan Babaei conceived and designed the experiments, analyzed the data, prepared figures and/or tables, and approved the final draft.
- Vladimir Simic conceived and designed the experiments, analyzed the data, prepared figures and/or tables, and approved the final draft.
- Erfan Babaee Tirkolaee conceived and designed the experiments, analyzed the data, prepared figures and/or tables, and approved the final draft.

### Data Availability
The data and code are available at GitHub and Zenodo:

- https://github.com/erfanmtl/variant-informed-decision-support-system

- Salehi, A., Babaei, A., Simic, V., & Babaee Tirkolaee, E. (2024). Amir27Salehi/variant-informed-decision-support-system: Initial Release (v1.0.0). Zenodo. https://doi.org/10.5281/zenodo.12804681.

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
