# Peer review of "A variant-informed decision support system for tackling COVID-19: a transfer learning and multi-attribute decision-making approach"

_PeerJ Computer Science, doi:10.7717/peerj-cs.2321_

## Round 0.1 · original submission · Major Revisions

I have received the review reports for your paper submitted to PeerJ Computer Science from the reviewers. According to the reports, I will recommend major revision to your paper. Please refer to the reviewers’ opinions to improve your paper. Please also write a revision note such that the reviewers can easily check whether their comments are fully addressed. We look forward to receiving your revised manuscript soon.

·

Basic reporting

The manuscript is written in clear, professional English and is generally easy to follow. The introduction provides a comprehensive overview of the COVID-19 pandemic's impact, the emergence of Variants of Concern (VOCs), and the necessity for an adaptive decision support system. The background and context are well-established, demonstrating the relevance and urgency of the research. The literature is well-referenced, citing recent and relevant studies, which strengthens the manuscript's foundation.

Request for correction:
(1) Raw data is mentioned, although its accessibility and detail should be explicitly confirmed. Can you cite and reference the source of your dataset?

(2) In line 189, Hasell et al., 2020, and Mathieu et al., 2021 are not referenced.

Experimental design

The research presents an original primary investigation within the scope of the journal. The research question is well-defined, relevant, and meaningful, addressing a significant gap in knowledge regarding decision support systems for managing COVID-19 variants. The use of Multi-Attribute Decision-Making (MADM) techniques and transfer learning to enhance neural network models is innovative and appropriate for the study's objectives.

The methods are described in sufficient detail to allow replication, covering data sources, preprocessing steps, model development, and evaluation criteria. The study adheres to high technical and ethical standards, including a comprehensive discussion of the data and models used.

Validity of the findings

Upon review of the provided code and the resulting outputs (https://github.com/erfanmtl/variant-informed-decision-support-system/blob/main/Code.ipynb), it is evident that the trained models have not successfully learned from the dataset. This issue critically undermines the validity of the results presented in the manuscript. The generated graphs indicate that the validation loss for all the variants ('Alpha', 'Beta', 'Gamma', 'Delta', 'Omicron') does not demonstrate any meaningful learning or generalization. All models exhibit signs of overfitting and perform poorly on the test data.

Detailed Observations:
1. Alpha Variant:
- The training process terminated early due to a lack of learning, as evidenced by the stagnation in the training metrics. There was no decrease in both the training and validation losses.

2. Beta Variant:
- The loss at the final epoch was higher than at the initial epoch, indicating that the model did not learn effectively from the dataset.

3. Gamma Variant:
- Similar to the Beta variant, the loss increased over time, showing no learning progression.

4. Delta Variant:
- Exhibits the same issues as the Beta and Gamma variants, with no effective learning observed.

5. Omicron Variant:
- Although slightly better, the results are still suboptimal and indicate insufficient learning.

It is important to note that the models' ability to generate "feature importance" does not inherently validate their performance. If the models fail to learn any discernible patterns from the dataset, the feature importance derived from these models may be incorrect and misleading.

Recommendations for Improvement:
To address these issues, it is recommended to rerun the models using different hyperparameters. It is crucial to ensure that the validation loss decreases consistently during the training process, indicating effective learning. Consider experimenting with the following:
- Adjusting the learning rate.
- Modifying the architecture of the neural networks.
- Implementing regularization techniques to mitigate overfitting.
- Increasing the dataset size or augmenting the data to improve generalization.

Revised models with effective learning may yield different feature importance and, consequently, different results than those presented in the current manuscript. It is imperative to ensure that the models are properly validated to provide reliable and accurate insights.

Additional comments

1. Evaluation Metrics:
- Provide a comprehensive set of evaluation metrics beyond just the loss.

2. Model Validation:
- Implement cross-validation techniques to ensure the models are not just fitting to a specific train-test split but are generalizing well across different subsets of the data.

3. Comparative Analysis:
- If possible, compare the performance of the proposed models with baseline models or existing methods in the literature. This comparison will help in understanding the relative efficacy of the new approach.

Addressing these points will significantly improve the manuscript's quality, providing a clearer and more reliable presentation of the research findings. This will also ensure that the conclusions drawn are well-supported and can be confidently referenced by other researchers and practitioners in the field.

Reviewer 2 ·

Basic reporting

See below

Experimental design

.

Validity of the findings

.

Additional comments

>> The language usage throughout this paper need to be improved, the author should do some proofreading on it.
>> Your abstract does not highlight the specifics of your research or findings. Rewrite the Abstract section to be more meaningful. I suggest to be Problem, Aim, Methods, Results, and Conclusion.
>> Introduction section can be extended to add the issues in the context of the existing work and how proposed algorithms/approach can be used to overcome this.
>> The problems of this work are not clearly stated. There is ambiguity in statement understanding.
>> Add main contributions list as points in the Introduction section.
>> Add the rest organization section at the end of the Introduction section.
>> More clarifications and highlighted about the research gabs in the related works section. I suggest to discuss the following studies:
- A robust framework for the selection of optimal COVID-19 mask based on aggregations of interval-valued multi-fuzzy hypersoft sets. Expert systems with applications
- MEF: multidimensional examination framework for prioritization of COVID-19 severe patients and promote precision medicine based on hybrid multi-criteria decision-making approaches. Bioengineering
- An Optimized Decision Support Model for COVID-19 Diagnostics Based on Complex Fuzzy Hypersoft Mapping. Mathematics
>> I feel that more explanation would be need on how the proposed method is performed.
>> How is the architecture of the Variant-Informed Decision Support System (VIDSS) structured to adapt dynamically to the unique characteristics of different Variants of Concern (VOCs)?
>> Can you explain the process of implementing transfer learning in VIDSS? How does it leverage insights from forecast models of previous VOCs to enhance predictions for future variants?
>> What specific Multi-Attribute Decision-Making (MADM) techniques are utilized in VIDSS, and how are they applied to assess a country’s performance relative to past states and other countries?
>> How does VIDSS integrate and analyze diverse data sources, such as population density, vaccination status, HDI, GDP per capita, and other socio-economic factors, to make informed decisions?
>> What performance metrics are used to evaluate the effectiveness of VIDSS, and how does it compare to traditional decision support systems in managing COVID-19?
>> How are distinct feature dynamics across different VOCs identified, and what role do factors like population density and vaccination status play in these dynamics?
>> What insights does VIDSS provide regarding the impact of vaccination on features like diabetes prevalence, cardiovascular death rate, and other health indicators post-vaccination?
>> What is the pioneering criterion introduced in VIDSS for refining country performance assessment, and how does it provide a nuanced perspective compared to traditional criteria?
>> How does VIDSS account for gender-specific distinctions, such as female smoker-related features, in its analysis and predictions?
>> Authors should add the parameters of the algorithms.
>> A comparison with state of art in the form of table should be added
>> Results need more explanations. Additional analysis is required at each experiment to show the its main purpose.
>> The Limitations of the proposed study need to be discussed before conclusion.
>> Rewrite the Conclusion section to be:
- You must more clearly highlight the theoretical and practical implications of your research
-Discuss research contributions.
-Indicate practical advantages (in at least one separate paragraph),
-discuss research limitations (at least one separate paragraph), and
-supply 2-3 solid and insightful future research suggestions.

Reviewer 3 ·

Basic reporting

All comments have been added in detail to the last section.

Experimental design

All comments have been added in detail to the last section.

Validity of the findings

All comments have been added in detail to the last section.

Additional comments

Review Report for PeerJ Computer Science
(A variant -informed decision support system for tackling COVID-19: A transfer learning and MADM approach)

1. Within the scope of the study, a variant-informed decision support system was developed using multi-attribute decision-making techniques.

2. In the introduction, the COVID-19 pandemic, artificial intelligence studies including variants of concerns in the literature, deficiencies in the literature and the contributions of this study to the literature are mentioned in detail and clearly.

3. It is understood that the dataset and amount used in the study are suitable and sufficient for the solution of the problem within the scope of the study. In addition, when the VIDSS framework is examined, it is observed that it has a certain level of originality.

4. When the number and description of the COVID-19 dataset's features are examined, it is seen that they are suitable for the scope of the study. Similarly, when the positive and negative criteria of multi-attribute decision-making are examined, it is observed that they are clearly expressed.

5. The results obtained, the observation of the effect of evaluation metrics and transfer learning increased the quality of the study.

6. Information about the neural network model and the algorithm should definitely be detailed. In addition, how the value/type of each of the model parameters such as learning rate, epoch, optimizer is determined should be explained more clearly. For example, how was the learning rate determined? How will any increase/decrease in the learning rate affect the result? Similarly, how was the epoch number determined and how will the increase/decrease in the epoch affect the result?

7. It is recommended to draw the basic block diagram of the model architecture. Indicate the reason for the activation function choice and compare it with the literature.

In conclusion, the study is important in terms of the subject discussed, but pay attention to the sections listed above.

---

## Round 0.2 · accepted · Accept

The authors have fully addressed the comments of the reviewers. I am happy to make a decision of acceptance to the paper.

·

Basic reporting

No comment

Experimental design

No comment

Validity of the findings

No comment

Reviewer 2 ·

Basic reporting

>> The authors have been addressed all my comments correctly. No more comments are required from my side. The current version can be published in the journal.

Experimental design

>> The authors have been addressed all my comments correctly. No more comments are required from my side. The current version can be published in the journal.

Validity of the findings

>> The authors have been addressed all my comments correctly. No more comments are required from my side. The current version can be published in the journal.

Additional comments

>> The authors have been addressed all my comments correctly. No more comments are required from my side. The current version can be published in the journal.

Reviewer 3 ·

Basic reporting

All comments have been added in detail to the last section.

Experimental design

All comments have been added in detail to the last section.

Validity of the findings

All comments have been added in detail to the last section.

Additional comments

Review Report for PeerJ Computer Science
(A variant-informed decision support system for tackling COVID-19: A transfer learning and multi-attribute decision-making approach)

Thank you for the revision. When the responses to the reviewer comments and the relevant changes in the paper are examined, although some are limited, they are generally appropriate. Therefore, I recommend that the paper be accepted. I wish the authors success in their future work. Best regards.